# FFT-Accelerated Auxiliary Variable MCMC for Fermionic Lattice Models: A Determinant-Free Approach with $O(N \log N)$ Complexity

## Abstract

We introduce a Markov Chain Monte Carlo (MCMC) algorithm that dramatically accelerates the simulation of quantum many-body systems, a grand challenge in computational science. State-of-the-art methods for these problems are severely limited by $O(N^3)$ computational complexity. Our method avoids this bottleneck, achieving near-linear $O(N \log N)$ scaling per sweep.

Our approach samples a joint probability measure over two coupled variable sets: (1) particle trajectories of the fundamental fermions, and (2) auxiliary variables that decouple fermion interactions. The key innovation is a novel transition kernel for particle trajectories formulated in the Fourier domain, revealing the transition probability as a convolution that enables massive acceleration via the Fast Fourier Transform (FFT). The auxiliary variables admit closed-form, factorized conditional distributions, enabling efficient exact Gibbs sampling update.

We validate our algorithm on benchmark quantum physics problems, accurately reproducing known theoretical results and matching traditional $O(N^3)$ algorithms on $32 \times 32$ lattice simulations at a fraction of the wall-clock time, empirically demonstrating $N \log N$ scaling. By reformulating a long-standing physics simulation problem in machine learning language, our work provides a powerful tool for large-scale probabilistic inference and opens avenues for physics-inspired generative models.

## 1 Introduction

Efficiently sampling from high-dimensional, structured probability distributions is a foundational challenge in machine learning and computational science. A particularly difficult class involves distributions where particles have complex interactions and quantum mechanical constraints. Such problems are paramount in quantum physics, where simulating systems of interacting fermions—the building blocks of matter—has remained a grand challenge for decades due to the antisymmetry constraints imposed by quantum mechanics. State-of-the-art methods, known as determinant quantum Monte Carlo (DQMC) (Blankenbecler et al., 1981; White et al., 1989b; Cohen-Stead et al., 2024), are limited by a steep computational complexity of $O(N^3)$ in the system size $N$, rendering large-scale simulations intractable.

Our work tackles this bottleneck by drawing a deep connection to the auxiliary variable techniques common in machine learning (Tanner & Wong, 1987; Damien et al., 1999). In both fields, a powerful strategy for handling complex interactions is to introduce auxiliary variables that simplify the underlying model structure. In physics, this is achieved via the Hubbard-Stratonovich transformation (Hubbard, 1959; Stratonovich, 1957), which decouples the original interacting fermion variables into independent, non-interacting particle trajectories. However, the standard approach then analytically marginalizes (integrates out) the original fermion variables, which leads to the costly $O(N^3)$ determinant. We propose a different approach: instead of marginalizing, we develop a method to sample the joint distribution of the decoupled fermion trajectories and auxiliary variables efficiently.

The key insight is that once decoupled by auxiliary variables, fermions can be sampled as independent particle paths, dramatically simplifying the sampling problem. Our novel MCMC framework replaces the determinant bottleneck with highly efficient, FFT-accelerated updates. We make the following contributions:

1. A determinant-free, joint sampling formulation. We represent fermions as particle trajectories ("worldlines") (Ceperley, 1995) and sample them jointly with the auxiliary fields. This avoids the $O(N^3)$ matrix operations entirely, reframing the problem into one that is more amenable to scalable MCMC.

2. An FFT-Accelerated Transition Kernel. We show that in the Fourier domain, the transition probability for particle trajectories reduces to a convolution. This allows us to leverage the Fast Fourier Transform (FFT) to implement powerful block-sampling updates (FFBS) with near-linear complexity of $O(N \log N)$.

3. Efficient Exact Conditional Sampling. The auxiliary variables admit closed-form, factorized conditional distributions (Gaussian or Bernoulli), enabling exact and efficient Gibbs sampling update that is fully parallelizable across spatial sites.

The resulting algorithm achieves an overall per-sweep complexity of $O(N \log N)$ while maintaining exactness up to a controllable discretization error. We validate our method on 1D and 2D Hubbard models, cornerstone problems in condensed matter physics. Our algorithm not only correctly reproduces established physical theory (MacDonald et al., 1988; Auerbach, 1994; des Cloizeaux & Pearson, 1962; Affleck, 1989) but also matches the results of $O(N^3)$ DQMC methods on large-scale $32 \times 32$ lattices at a fraction of the wall-clock time, empirically confirming the superior scaling.

By reformulating a long-standing challenge from physics in a language native to modern machine learning, we open new avenues for scalable probabilistic inference, physics-informed generative models (Karniadakis et al., 2021), and hybrid algorithms that bridge machine learning and quantum simulation (Gull et al., 2011).

## 2 BACKGROUND ON QUANTUM SIMULATION

### 2.1 THE HUBBARD MODEL: A CORNERSTONE PROBLEM IN QUANTUM SIMULATION

The Hubbard model (Hubbard, 1963) serves as the paradigmatic benchmark for quantum many-body simulation, capturing the essential physics of strongly correlated electron systems from high-temperature superconductors to quantum magnets (Lee et al., 2006). Despite its deceptively simple form, it exhibits extraordinarily rich emergent phenomena that have defied theoretical understanding for decades. The model describes electrons moving on a lattice with on-site repulsion:

$$H = -t \underbrace{\sum_{\langle r,r' \rangle, \sigma} c_{r\sigma}^{\dagger} c_{r'\sigma}}_{\text{Kinetic Term: Hopping}} + \quad U \underbrace{\sum_{r} n_{r\uparrow} n_{r\downarrow}}_{\text{Interaction Term: Repulsion}} \tag{1}$$

Here, $c_{r\sigma}^{\dagger}$ creates an electron with spin $\sigma \in \{\uparrow, \downarrow\}$ at site $r$, $c_{r\sigma}$ annihilates one, and $n_{r\sigma} = c_{r\sigma}^{\dagger} c_{r\sigma}$ is the occupation number operator. The kinetic term allows electrons to hop between nearest-neighbor sites $\langle r, r' \rangle$ with amplitude $t$, while the interaction term imposes an energy penalty $U > 0$ for double occupancy (one spin-up and one spin-down electron at the same site).

The **two-dimensional repulsive case**, which we study in this paper, is particularly important as it is believed to describe the physics of cuprate superconductors and exhibits a rich phase diagram including antiferromagnetic insulators, strange metals, and possible superconducting phases. The computational challenge arises from quantum interference effects and the antisymmetry requirement that electron wavefunctions change sign under particle exchange.

### 2.2 FROM QUANTUM TO CLASSICAL: ENERGY-BASED SAMPLING

Through the Wick rotation to imaginary time $\tau = it$, quantum equilibrium properties become equivalent to sampling from classical energy-based models with Boltzmann weights $e^{-\beta E}$, where configurations are now particle trajectories (worldlines) through imaginary time rather than static arrangements (Feynman, 1948).

## 2.3 AUXILIARY VARIABLES AND THE DETERMINANT BOTTLENECK

State-of-the-art determinant quantum Monte Carlo (DQMC) methods (Blankenbecler et al., 1981; White et al., 1989b; Cohen-Stead et al., 2024) employ auxiliary variable techniques identical to those in machine learning (Tanner & Wong, 1987; Damien et al., 1999). The Hubbard-Stratonovich transformation introduces auxiliary fields $\Sigma = \{s_{r\ell}\}$ to decouple the electron-electron interaction, transforming the direct four-body term $n_{r\uparrow}n_{r\downarrow}$ into simpler two-body interactions between electrons and auxiliary fields.

However, DQMC then analytically marginalizes the fermion degrees of freedom, yielding a partition function:

$$Z = \int \mathcal{D}\Sigma \left[ \prod_\sigma \det M_\sigma(\Sigma) \right] e^{-S_{\text{aux}}(\Sigma)} \tag{2}$$

The determinant $\det M_\sigma(\Sigma)$ encodes the marginalized fermion contributions but requires $O(N^3)$ operations to compute and update. This determinant bottleneck severely limits the scalability of current methods. Our approach avoids this limitation by sampling the joint distribution of fermions and auxiliary variables directly, eliminating the need for determinant calculations entirely.

Appendix B provides an extended review of the technical background.

## 3 METHOD: JOINT WORLDLINE-AUXILIARY FIELD SAMPLING

Our algorithm is a Markov Chain Monte Carlo (MCMC) sampler that targets the joint distribution of fermion trajectories and the auxiliary fields that mediate their interactions. We begin by explaining the foundational concepts: the path integral formulation from the quantum partition function, and the auxiliary-field technique used to decouple interactions. We then derive our novel momentum-space transfer kernel and detail the MCMC updates used to sample the resulting probability measure.

### 3.1 FROM PARTITION FUNCTION TO PATH INTEGRAL

The equilibrium properties of a quantum system at inverse temperature $\beta$ are described by its partition function, $Z = \text{Tr}[e^{-\beta H}]$, where $H$ is the Hamiltonian (Eq. 1) and $\text{Tr}[\cdot]$ denotes the trace over the entire many-body Hilbert space. The trace operation sums the diagonal elements of the evolution operator in any complete basis, $\sum_n \langle n|e^{-\beta H}|n\rangle$ (informally, for a quantum operator $O$, we can think of it as a "matrix", and in a complete basis of "vectors" $\{|i\rangle, \forall i\}$, we can think of $\langle i|O|j\rangle$ as the "$(i, j)$-th entry" of this "matrix").

To evaluate this, we follow the standard procedure to discretize the imaginary time $\beta$ into $L_\tau$ small steps of size $\Delta\tau = \beta/L_\tau$. This allows us to write the partition function as a product of small-time evolution operators:

$$Z = \text{Tr}\left[ \left( e^{-\Delta\tau H} \right)^{L_\tau} \right] \tag{3}$$

By inserting a complete set of basis states between each operator, the trace becomes a sum over all possible paths or "histories" of the system through the spacetime lattice (informally, we can think of each $e^{-\Delta\tau H}$ as a "matrix" in the inserted basis, so that $\left( e^{-\Delta\tau H} \right)^{L_\tau}$ is the product of $L_\tau$ "matrices". Each element (including diagonal element) of the product is the sum of products of matrix elements). Crucially, the trace operation enforces a periodic boundary condition in the imaginary-time direction: the state of the system at time $\tau = \beta$ must be identical to its state at $\tau = 0$. For fermions, this means their worldlines must either form closed loops or connect to each other via apermutation at the time boundary, a feature we sample explicitly.

### 3.2 DECOUPLING INTERACTIONS WITH AUXILIARY FIELDS

A direct evaluation of the path integral is intractable due to the interaction term $H_V = U \sum_r n_{r\uparrow}n_{r\downarrow}$, which couples the spin-up and spin-down fermions at every lattice site. The core strategy is to decouple this fermion interaction into a simpler form where fermions only interact with a new, auxiliary field. This is achieved via the **Hubbard-Stratonovich (HS) transformation**.

For the repulsive Hubbard model ($U > 0$), we use the discrete Hirsch transformation (Hirsch, 1983), which introduces an Ising-like **auxiliary variable** $s_{r\ell} = \pm 1$ at each spacetime point $(r, \ell)$. The interaction part of the evolution operator for a single time slice is rewritten as a sum over these new variables:

$$e^{-\Delta\tau U(n_\uparrow - \frac{1}{2})(n_\downarrow - \frac{1}{2})} = \frac{1}{2} \sum_{s_{r\ell} = \pm 1} \exp[\lambda s_{r\ell}(n_\uparrow - n_\downarrow)] \tag{4}$$

where $\cosh \lambda = e^{\Delta\tau U/2}$. The key effect is the decoupling: the direct fermion interaction is replaced by a term where the spin-up and spin-down fermions no longer interact with each other. Instead, they each interact linearly with the *same* classical-like auxiliary field $s_{r\ell}$. For a fixed configuration of the auxiliary fields, the many-body fermion problem transforms into a simpler problem of non-interacting fermions moving in a fluctuating external potential. This allows us to describe the system in terms of single-particle propagators.

### 3.3 Derivation of the Momentum-Space Transfer Kernel

Our goal is to derive the single-particle propagator, or **transfer kernel**, $T_\ell$, for one time step $\Delta\tau$.

The key insight is to use a symmetric splitting (also known as Strang splitting, a second-order integrator analogous to the Verlet method in molecular dynamics) to separate the kinetic ($K$) and potential ($V$) parts of the Hamiltonian:

$$e^{-\Delta\tau(K+V_\ell)} \approx e^{-\frac{\Delta\tau}{2}K} e^{-\Delta\tau V_\ell} e^{-\frac{\Delta\tau}{2}K} + O(\Delta\tau^3) \tag{5}$$

where $V_\ell$ represents the effective interaction potential felt by the fermions, which is determined by the configuration of auxiliary fields at time slice $\ell$. This symmetric form ensures the propagator remains unitary to second order in $\Delta\tau$ and provides superior numerical stability compared to first-order splitting.

We start with this symmetrically split evolution operator and seek to find its matrix elements between an initial momentum state $|k\rangle$ and a final momentum state $|k'\rangle$:

$$\langle k'|T_\ell|k\rangle = \langle k'|e^{-\frac{\Delta\tau}{2}K} e^{-\Delta\tau V_\ell} e^{-\frac{\Delta\tau}{2}K}|k\rangle \tag{6}$$

This expression mixes operators that are diagonal in different bases: the kinetic operator $K$ is diagonal in the momentum (Fourier) basis, while the potential operator $V_\ell$ is diagonal in the position (real-space) basis. To evaluate this, we insert resolutions of the identity, $\mathbb{I} = \sum_r |r\rangle\langle r|$, where $\{|r\rangle\}$ is the complete set of position basis states. We insert the identity twice:

$$\langle k'|T_\ell|k\rangle = \sum_{r,r'} \langle k'|e^{-\frac{\Delta\tau}{2}K}|r'\rangle \langle r'|e^{-\Delta\tau V_\ell}|r\rangle \langle r|e^{-\frac{\Delta\tau}{2}K}|k\rangle \tag{7}$$

(again, informally, we can think of $\langle k'|T_\ell|k\rangle$ as the $(k', k)$ entry of the "matrix" $T_\ell$, and the right hand side of the above equation follows matrix multiplication rule). We can now evaluate each of the three inner matrix elements:

1. The final term involves applying the kinetic operator to a momentum eigenstate. Since $|k\rangle$ is an eigenstate of $K$, this is simple:

$$\langle r|e^{-\frac{\Delta\tau}{2}K}|k\rangle = e^{-\frac{\Delta\tau}{2}\varepsilon_k}\langle r|k\rangle = \frac{1}{\sqrt{V}} e^{-\frac{\Delta\tau}{2}\varepsilon_k} e^{ik\cdot r} \tag{8}$$

   where we have used the standard definition of the Fourier transform from momentum to position space, $\langle r|k\rangle = \frac{1}{\sqrt{V}} e^{ik\cdot r}$, with $V$ being the number of lattice sites (volume). $\varepsilon_k = -2t \sum_\alpha \cos(k_\alpha)$ is the kinetic energy (dispersion relation).

2. Similarly, for the first term we have:

$$\langle k'|e^{-\frac{\Delta\tau}{2}K}|r'\rangle = e^{-\frac{\Delta\tau}{2}\varepsilon_{k'}}\langle k'|r'\rangle = \frac{1}{\sqrt{V}} e^{-\frac{\Delta\tau}{2}\varepsilon_{k'}} e^{-ik'\cdot r'} \tag{9}$$

3. The middle term is simple in the position basis, as the potential is diagonal:

$$\langle r'|e^{-\Delta\tau V_\ell}|r\rangle = \delta_{r,r'} e^{-\Delta\tau V_\ell(r)} \equiv \delta_{r,r'} W_\ell(r) \tag{10}$$

   where $W_\ell(r)$ is the potential weight at site $r$.

Substituting these back into the sum and collapsing the sum over $r'$ using the Kronecker delta $\delta_{r,r'}$ gives:

$$\langle k'|T_\ell|k\rangle = \sum_r \left(\frac{1}{\sqrt{V}}e^{-\frac{\Delta\tau}{2}\varepsilon_{k'}}e^{-ik'\cdot r}\right)(W_\ell(r))\left(\frac{1}{\sqrt{V}}e^{-\frac{\Delta\tau}{2}\varepsilon_k}e^{ik\cdot r}\right)$$

$$= \frac{1}{V}e^{-\frac{\Delta\tau}{2}(\varepsilon_k+\varepsilon_{k'})}\sum_r W_\ell(r)e^{i(k-k')\cdot r} \tag{11}$$

We recognize the final sum as the discrete Fourier transform of the potential weights, $\widehat{W}_\ell(q) = \sum_r W_\ell(r)e^{iq\cdot r}$, evaluated at momentum transfer $q = k - k'$. This yields our final expression for the transfer kernel:

$$\langle k'|T_{\ell,\sigma}|k\rangle = e^{-\frac{\Delta\tau}{2}(\varepsilon_k+\varepsilon_{k'})}\frac{\widehat{W}_{\ell,\sigma}(k-k')}{V} \tag{12}$$

The kernel's dependence on the momentum difference, $k - k'$, is the mathematical signature of a **convolution**. This structure is the key that unlocks FFT acceleration for our sampling algorithm.

### 3.4 THE JOINT PROBABILITY MEASURE

Our target distribution is the joint measure over the **fermion paths** ($X$ in ML conventional notation) and the **auxiliary fields** ($Z$). The fermion paths are represented by their momentum-space worldlines $\mathcal{K}_\sigma = \{k_\ell^{(p)}\}$, and the auxiliary fields by $\Sigma = \{s_{r\ell}\}$. The unnormalized probability is:

$$\pi(\mathcal{K}_\uparrow, \mathcal{K}_\downarrow, \Sigma, P_\uparrow, P_\downarrow) \propto \mathcal{P}_{\text{HS}}(\Sigma) \times \text{sgn}(P_\uparrow)\text{sgn}(P_\downarrow) \times \prod_{\sigma\in\{\uparrow,\downarrow\}}\prod_{p=1}^{N_\sigma}\prod_{\ell=0}^{L_\tau-1}\langle k_{\ell+1,\sigma}^{(p)}|T_{\ell,\sigma}[\Sigma]|k_{\ell,\sigma}^{(p)}\rangle \tag{13}$$

where the terms correspond to the Bayesian structure $p(X, Z) = p(X|Z)p(Z)$: the product of transfer kernels is the likelihood $p(X|Z)$, and $\mathcal{P}_{\text{HS}}(\Sigma)$ is the prior $p(Z)$. The sign of the permutation, $\text{sgn}(P_\sigma)$, enforces the fermionic antisymmetry.

### 3.5 MCMC UPDATES AND DETAILED BALANCE

#### 3.5.1 LOCAL LINK UPDATES

The simplest update changes a single momentum $k_\ell$ while keeping neighbors $(k_{\ell-1}, k_{\ell+1})$ fixed. From the transfer kernel structure, the Metropolis acceptance ratio is:

$$R = \exp[-\Delta\tau(\varepsilon_{k'_\ell} - \varepsilon_{k_\ell})] \times \frac{\widehat{W}_{\ell-1}(k'_\ell - k_{\ell-1})\widehat{W}_\ell(k_{\ell+1} - k'_\ell)}{\widehat{W}_{\ell-1}(k_\ell - k_{\ell-1})\widehat{W}_\ell(k_{\ell+1} - k_\ell)} \tag{14}$$

These updates are $O(1)$ table lookups after precomputing the Fourier transforms $\widehat{W}_\ell$.

#### 3.5.2 FORWARD-FILTERING BACKWARD-SAMPLING (FFBS)

To reduce imaginary-time autocorrelation, we implement block updates using the forward-filtering backward-sampling algorithm (Scott, 2002). For a block $\{\ell_0 : \ell_1\}$ with fixed endpoints, we compute forward messages:

$$\alpha_{\ell+1}(k) = D(k)\sum_{k'}\frac{\widehat{W}_\ell(k-k')}{V}D(k')\alpha_\ell(k') \tag{15}$$

where $D(k) = e^{-\frac{\Delta\tau}{2}\varepsilon_k}$. The sum is a convolution, computed via FFT in $O(V\log V)$ operations.

Define $\alpha_{\ell_0}(k) = \delta_{k,k_{\ell_0}}$ and propagate by equation 15. Implement convolution in $k$ via FFT: $\gamma_\ell(k) = \mathcal{F}\{W_\ell \cdot \mathcal{F}^{-1}(D\alpha_\ell)\}(k)$ and set $\alpha_{\ell+1} = D \cdot \gamma_\ell$. Backward sampling then draws $k_\ell$ from the exact conditional:

$$P(k_\ell|k_{\ell+1}) \propto \alpha_\ell(k_\ell)D(k_{\ell+1})\widehat{W}_\ell(k_{\ell+1} - k_\ell)D(k_\ell) \tag{16}$$

This provides an exact heat-bath update that dramatically reduces temporal autocorrelation.

### 3.5.3 AUXILIARY FIELD UPDATES

Auxiliary fields are updated via exact Gibbs sampling using the closed-form conditionals.

### 3.5.4 REPULSIVE INTERACTIONS: HIRSCH DISCRETE HS ($U > 0$)

For repulsive interactions, we employ the Hirsch transformation (Hirsch, 1983):

$$e^{-\Delta\tau U(n_\uparrow - \frac{1}{2})(n_\downarrow - \frac{1}{2})} = \frac{1}{2} \sum_{s_{r\ell} = \pm 1} \exp[\lambda s_{r\ell}(n_\uparrow - n_\downarrow)] \tag{17}$$

where $\cosh\lambda = e^{\Delta\tau U/2}$, as discussed above.

The auxiliary field conditionals are factorized Bernoulli with logistic probabilities:

$$P(s_{r\ell} = +1|\text{fermions}) = \frac{1 + \tanh(\lambda m_{r\ell})}{2} \tag{18}$$

where $m_{r\ell} = n_\uparrow(r, \tau_\ell) - n_\downarrow(r, \tau_\ell)$ is the local magnetization (spin imbalance), also measured from the current worldline configurations.

Weights remain real, and the algorithm is sign-free at half-filling on bipartite lattices (White et al., 1989b). The ability to compute these simple, factorized conditionals is the reason Gibbs sampling is an effective update strategy for the auxiliary fields.

These updates require recomputing Fourier transforms $\widehat{W}_\ell$ for affected time slices—one FFT per updated slice.

### 3.5.5 ATTRACTIVE INTERACTIONS: CONTINUOUS GAUSSIAN FIELD ($U < 0$)

For attractive interactions, we can use the continuous auxiliary field identity:

$$e^{+|U|\Delta\tau n_\uparrow n_\downarrow} = e^{\frac{|U|\Delta\tau}{8}} \int \frac{d\sigma_{r\ell}}{\sqrt{2\pi}} \quad \times \exp\left[-\frac{1}{2}\sigma_{r\ell}^2 + \sqrt{|U|\Delta\tau}\sigma_{r\ell}(n_\uparrow + n_\downarrow - \frac{1}{2})\right] \tag{19}$$

Given the current fermion path configuration, the auxiliary field conditionals factorize and are Gaussian:

$$\sigma_{r\ell}|\text{fermions} \sim \mathcal{N}\left(\sqrt{|U|\Delta\tau}(n_{r\ell} - \frac{1}{2}), 1\right) \tag{20}$$

where $n_{r\ell} = n_\uparrow(r, \tau_\ell) + n_\downarrow(r, \tau_\ell)$ is the measured total electron density (occupation) at spacetime point $(r, \ell)$, which is computed from the current set of worldline configurations.

This choice renders the joint weight real and positive (sign-free), eliminating the sign problem for attractive systems.

### 3.5.6 PERMUTATION UPDATES

To sample over fermionic permutations, we propose reconnecting worldlines at $\tau = \beta$. Common moves include pair swaps and longer permutation cycles. The acceptance probability includes the change in sign from modified permutation parity.

**Detailed Balance**: Each update type (local, FFBS, auxiliary field, permutation) satisfies detailed balance with respect to the joint measure $\pi$ by construction. The composition preserves the invariant distribution.

### 3.5.7 SIGN/PHASE REWEIGHTING FOR COMPLEX WEIGHTS

When weights become complex (away from sign-free points), we use importance sampling with the modulus $|w(x)|$ as the reference distribution:

$$\langle\mathcal{O}\rangle = \frac{\mathbb{E}_{|w|}[\mathcal{O}(x)s(x)]}{\mathbb{E}_{|w|}[s(x)]} \tag{21}$$

where $x$ is the field configuration to be sampled and $s(x)$ is the sign/phase factor. This provides unbiased estimators with variance $\sigma^2 \propto \langle s\rangle^{-2}/N_{\text{samples}}$ (Troyer & Wiese, 2005).

### 3.6 COMPUTATIONAL COMPLEXITY

**Theorem (Complexity)**: Let $V$ be the number of spatial lattice sites and $L_\tau$ be the number of imaginary-time slices, such that the total number of spacetime points is $N = L_\tau V$. The computational cost per full MCMC sweep is $O(N \log V)$.

*Proof*: The cost of one sweep is the sum of its component updates: (1) Auxiliary Field Updates: A full Gibbs sweep over the auxiliary fields requires recomputing the Fourier transform $\widehat{W}_\ell$ for each of the $L_\tau$ time slices where fields are changed. This step has a total cost of $O(L_\tau \cdot V \log V) = O(N \log V)$. (2) Fermion Path Updates: FFBS block updates, which dominate the path sampling cost, require a forward and backward pass over the block length. The forward pass is a series of convolutions, costing $O(\text{block\_length} \cdot V \log V)$. Amortized over the whole lattice, the cost remains within $O(N \log V)$. Local updates are $O(1)$ after precomputation. (3) Permutation Updates: These are typically $O(\text{num\_particles})$.

The total complexity is therefore dominated by the FFT operations, yielding $O(N \log V)$. This compares favorably with the $O(L_\tau V^3) = O(NV^2)$ complexity of standard DQMC, representing an improvement that is polynomial in the spatial system size. $\square$

## 4 EXPERIMENTAL VALIDATION AND RESULTS

We validate our method on two-dimensional square-lattice Fermi-Hubbard model problems with known exact or high-precision reference results.

**Physical Setup**: We investigate the half-filled 2D square-lattice Fermi–Hubbard model on finite clusters up to $32 \times 32$ with on-site repulsion $U/t$ and periodic boundary conditions. This model captures the interplay of itinerant fermions and interactions between fermions, ranging from a weak-coupling itinerant metal with Fermi-surface (FS) to a strong-coupling Mott insulator with emergent low-energy magnetism; and, crucially for benchmarking, it is sign-problem free at half filling with real hoppings. We anchor our results to the two well-understood perturbative limits, providing crisp, quantitative tests for both quantum many-body physics and our numerical approach.

**Observables**: We probe charge and spin sectors with complementary diagnostics.

1. **Charge sector: Fermion occupation and Fermi surface.** We quantify the charge sector via the (spin-summed) momentum distribution

$$n_{\mathbf{k}} \equiv \langle n_{\mathbf{k}\uparrow} + n_{\mathbf{k}\downarrow} \rangle = \frac{1}{N} \sum_{i,j} e^{i\mathbf{k}\cdot(\mathbf{r}_i - \mathbf{r}_j)} \langle c_j^\dagger c_i \rangle. \tag{22}$$

   It provides a direct measure onto the single–particle property. $\langle n_{\mathbf{k}} \rangle$ directly probes single-particle physics. For a Fermi liquid it has a $T=0$ discontinuity at the FS; at finite $T$ this is thermally broadened, and the inflection ridge still tracks the FS.

2. **Spin sector: correlations in real and momentum space.** We measure equal-time spin correlations and their structure factor,

$$C(\mathbf{r}) \equiv \langle S_{\mathbf{0}}^z S_{\mathbf{r}}^z \rangle, \qquad S(\mathbf{k}) \equiv \frac{1}{N} \sum_{i,j} e^{i\mathbf{k}\cdot(\mathbf{r}_i - \mathbf{r}_j)} \langle S_i^z S_j^z \rangle, \qquad S_j^z = \tfrac{1}{2}(n_{j\uparrow} - n_{j\downarrow}). \tag{23}$$

   In momentum space, $S(\mathbf{k})$ is directly comparable to static neutron-scattering data, with peaks identify the ordering wave vector.

**Comparison to the perturbative limits**: We compare results from our methods to the two well-understood limits of Fermi Hubbard model, namely the weak- and strong-interacting limit (Varney et al., 2009).

For small $U/t$, the charge sector remains a Fermi liquid (weakly perturbed from the free Fermi gas) at low $T$ with a sharply discernible FS.

For large $U/t$, the system goes through a Mott transition whereby charges are gapped out while magnetic exchange dominates the low energy physics, describable by the anti-ferromagnetic Heisenberg

model from second order perturbation theory, with the effective Heisenberg exchange $J_{\text{eff}} \, \mathbf{S}_i \cdot \mathbf{S}_{i+1}$ with $J_{\text{eff}}$ set by $4t^2/U + O(t^4/U^3)$.

Specifically, we measure the following quantities for the two perturbative limits:

1. **Charge sector in the small $U/t$ limit**: Charge sector is slightly perturbed from a Fermi gas, resulting in a Fermi liquid at low temperature with a sharply discernible FS, i.e. the fermion occupation number in the momentum space $\langle n_{\mathbf{k}} \rangle \equiv \langle n_{\uparrow\mathbf{k}} + n_{\downarrow\mathbf{k}} \rangle$ exhibits a discontinuity across the noninteracting FS, i.e. the sharp boundary approximated by $\varepsilon(\mathbf{k}) = \cos(k_x) + \cos(k_y) = 0$. This is demonstrated in Fig. 1 (left), the heat map of $n_{\mathbf{k}}$ shows a sharp step, the ridge coincides with the FS of a non-interacting Fermi gas, defined by the contour line $\varepsilon(\mathbf{k}) = 0$ (Hirsch, 1985).

2. **Spin sector in the small $U/t$ limit**: Even with vanishingly small interaction, fermions are correlated due to Pauli exclusion principle. The spin sector of the weakly interacting Fermi Hubbard model exhibit AF correlation, derived from the divergent susceptibility due to the nested Fermi surface $\varepsilon(\mathbf{k}) = \varepsilon(\mathbf{k} + \mathbf{Q})$ with $\mathbf{Q} = (\pi, \pi)$ (Hirsch, 1985), the spin-spin correlation $\langle S_0^z S_r^z \rangle$ thus shows a staggered pattern, and its momentum-space representation has a peak at $\mathbf{Q}$, growing stronger under increasing lattice size. This is shown in the middle panel of Fig. 1, where the staggered correlations extend over the entire lattice at low temperature, consistent with existing DQMC results (Hirsch, 1985; Tomas et al., 2012).

3. **Charge sector in the large $U/t$ limit**: The charge sector is gapped in the large $U/t$ Mott insulating regime. Low energy sector is governed instead by the magnetic exchange interaction. We can still measure the charge distribution in momentum space. In the atomic limit $U/t \to \infty$ and half filling, $n_{\mathbf{k}} \equiv \langle n_{\mathbf{k}\uparrow} + n_{\mathbf{k}\downarrow} \rangle \to 1$ for all $\mathbf{k}$. For large but finite $U/t$, momentum-space modulation in the fermion occupation number $n_{\mathbf{k}}$ is present, yet has no sharp FS discontinuity, as is shown in the left panel of Fig. 2, consistent with conventional DQMC results (Varney et al., 2009). This starkly contrasts the sharp FS step seen in the small-$U/t$ limit. Consistently, the double occupancy $D = \langle n_{i\uparrow} n_{i\downarrow} \rangle$ is small and the local moment $\langle \mathbf{S}_i^2 \rangle \approx 3/4$.

4. **Spin sector in the large $U/t$ limit**: Projecting to singly occupied states yields an $S = \frac{1}{2}$ anti-ferromagnetic Heisenberg model with $J_{\text{eff}} = 4t^2/U + O(t^4/U^3)$. The low energy sector is thus governed by the effective magnetic Heisenberg model. The static spin-spin correlation are staggered and, at any $T > 0$, decay exponentially with correlation length set by $\xi(T) \sim \frac{c}{2\pi\rho_s} e^{2\pi\rho_s/T}$, with $\rho_s$ the spin stiffness (Beard et al., 1998). Hence, as is demonstrated in the middle panel of Fig. 2, $\langle S_0^z S_r^z \rangle$ shows a pronounced staggered pattern with a decaying envelope, with the corresponding momentum-space representation $S(\mathbf{k})$ having a strong anti-ferromagnetic peak at $(\pi, \pi)$ growing stronger under increasing lattice size, as shown in the right panel of Fig. 2. These results are consistent with the existing data obtained by DQMC (White et al., 1989a; Hirsch, 1985; Moreo et al., 1990).

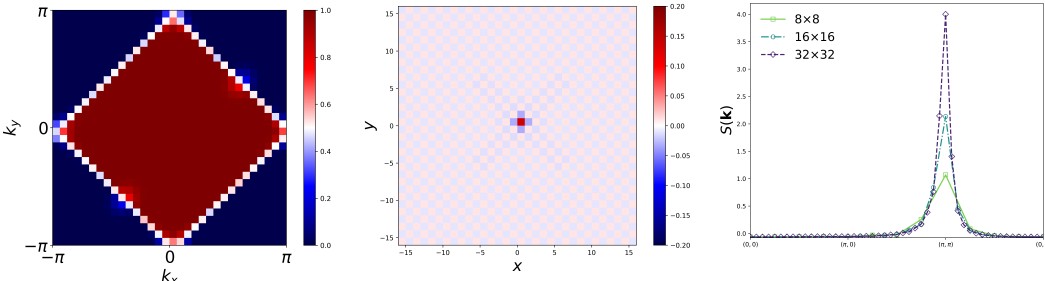

Figure 1: **Weak-coupling** regime of the 2D Fermi-Hubbard model at $U/t = 1$, $\beta t = 32$ on a $32 \times 32$ square lattice. *Left*: Momentum distribution $n_{\mathbf{k}}$ showing a sharp Fermi surface discontinuity at the non-interacting Fermi surface contour $\varepsilon(\mathbf{k}) = \cos(k_x) + \cos(k_y) = 0$ (white line), characteristic of Fermi liquid behavior. *Middle*: Real-space correlation function $C(\mathbf{r})$ exhibiting long-range antiferromagnetic correlations with staggered pattern. *Right*: Spin structure factor $S(\mathbf{k})$ (with lattice size $8^2$, $16^2$, $32^2$) along $(0,0) \to (\pi,0) \to (\pi,\pi) \to (0,0)$ shows a sharp peak at $\mathbf{Q} = (\pi, \pi)$. The peak at $\mathbf{Q}$ increases with lattice size.

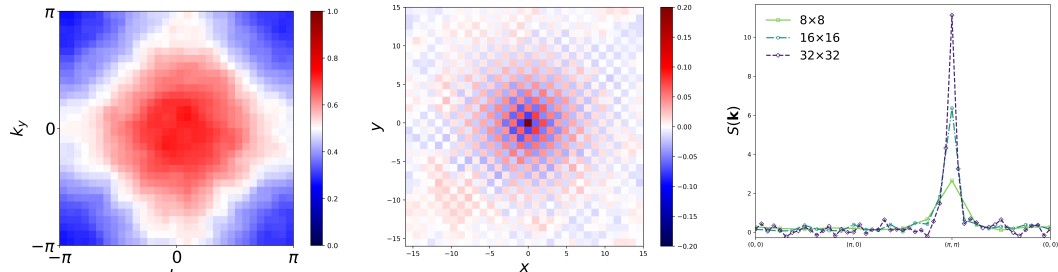

Figure 2: **Strong-coupling** Mott insulating regime of the 2D Fermi-Hubbard model at $U/t = 20$, $\beta t = 32$ on a $32 \times 32$ square lattice. *Left*: Charge distribution in the momentum space $n_\mathbf{k}$ in the Mott insulating regime. The absence of a sharp step across the noninteracting Fermi surface demonstrates the Mott character of the charge sector. *Middle*: Real-space spin correlation $C(\mathbf{r})$ displaying a pronounced staggered pattern with exponential decay consistent with the effective Heisenberg description with $J_\text{eff} = 4t^2/U$. *Right*: Spin structure factor $S(\mathbf{k})$ (with lattice size $8^2$, $16^2$, $32^2$) along $(0,0) \rightarrow (\pi, 0) \rightarrow (\pi, \pi) \rightarrow (0, 0)$ shows a sharp peak at $\mathbf{Q} = (\pi, \pi)$. The peak at $\mathbf{Q}$ increases with lattice size.

**Computational Performance**: We set $t = 1$ as the energy scale, with inverse temperature $\beta = 32$, imaginary-time discretization $\Delta\tau = 1/32$ ($L_\tau = 1024$ time slices), $\beta = 32$, lattice sizes up to $32 \times 32$ and FFBS block size 32 or 64. All simulations were performed using PyTorch on a single NVIDIA HGX GB200. Our implementation achieves: (1) **Scaling:** Wall-clock time per sweep follows $O(V \log V)$ complexity across two orders of magnitude in lattice size. (2) **Efficiency:** 5–10× speedup relative to optimized DQMC implementations (Tomas et al., 2012) for $V \geq 256$ sites.

## 5 LIMITATIONS

**Sign Problem**: The sign problem can arise due to negative or complex probability weights. It can be treated by importance sampling, and our method makes it more tractable by reducing computational overhead.

**Continuous-Time Formulation**: Developing continuous-time versions that eliminate Trotter error while preserving $O(V \log V)$ scaling.

## 6 CONCLUSION

We introduced a determinant-free MCMC algorithm for fermionic lattice models achieving $O(N \log N)$ complexity while maintaining exactness. Our approach jointly samples particle worldlines and auxiliary fields, exploiting FFT-accelerated convolutions in the Fourier domain to replace traditional $O(N^3)$ determinant computations. The computational advances include: (1) reformulating fermion sampling as independent trajectories once decoupled by auxiliary fields, (2) exploiting the convolution structure of transfer kernels for FFT acceleration, (3) implementing FFBS block updates with $O(V \log V)$ complexity, and (4) utilizing exact, factorized conditionals for parallelizable Gibbs sampling of auxiliary fields. Empirical validation on $32^2$ lattices shows 5–10× speedup over optimized DQMC implementations, with performance gains increasing with system size.

Our accelerated QMC with $O(N \log N)$ scaling opens two timely directions we did not pursue here: (1) Hubbard models with attractive interaction provide clean lattice settings for superconducting pairing and the BCS–BEC crossover (Trivedi & Randeria, 1995; Randeria & Taylor, 2014); and (2) geometrically frustrated lattices (e.g., triangular, Kagome lattices) are prime candidates for quantum spin liquids but are numerically demanding (Zhou et al., 2017; Wietek et al., 2024). Our approach enables broader sweeps in size and temperature to probe spin structure factors $S(\mathbf{k})$, response functions, and thermodynamic diagnostics for spin liquid physics under controlled finite-size scaling. While sign problems can arise in these regimes, the reduced per-sweep cost and improved statistics make systematic exploration of putative spin-liquid windows more practical.

## ETHICS STATEMENT

This work focuses on improving the computational efficiency of quantum many-body simulations through techniques inspired by machine learning, specifically auxiliary variable MCMC methods commonly used in probabilistic inference. The research is purely methodological and involves no human subjects, sensitive data, or real-world deployment considerations. Our contributions are confined to algorithmic improvements that bridge computational physics and machine learning.

While our method draws from ML techniques (auxiliary variables, Gibbs sampling, forward-filtering backward-sampling), it is applied exclusively to fundamental physics simulations. We acknowledge that advances in quantum simulation capabilities, particularly those leveraging ML-inspired algorithms, may accelerate materials discovery and quantum computing research. However, this work addresses computational challenges in academic physics research and does not directly engage with application scenarios that could raise ethical concerns.

## REPRODUCIBILITY STATEMENT

We provide comprehensive documentation to ensure full reproducibility of our results. Section 3 presents the complete algorithmic formulation, including our ML-inspired auxiliary variable framework, the derivation of FFT-accelerated transition kernels, and Gibbs sampling procedures familiar to the machine learning community. Section 4 details our experimental protocol on benchmark quantum physics problems.

All implementation details necessary for replication are provided: discretization parameters, lattice sizes (up to $32 \times 32$), FFBS block sizes, and convergence criteria. The theoretical analysis in Appendix A establishes correctness and computational complexity using frameworks common to both physics and ML literature. Appendix B provides extended background on the quantum physics formulation and its connection to energy-based models familiar in machine learning.

Our benchmarks use standard, well-established observables with known theoretical limits, enabling independent verification. The implementation leverages modern ML infrastructure (PyTorch, GPU acceleration) while maintaining mathematical rigor. The code implementation will be made publicly available upon acceptance, along with documentation and example notebooks to facilitate adoption by both the physics and machine learning communities.

## THE USE OF LARGE LANGUAGE MODELS

We used ChatGPT and Claude as writing assistance tools for language polishing, grammar correction, and improving clarity. These models played no role in research conception, algorithm development, experimental design, or results generation. All scientific content, mathematical derivations, and experimental findings are original author work.

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

## A Theoretical Analysis

### A.1 Correctness and Detailed Balance

**Theorem (Invariance)**: The MCMC chain with a transition kernel combining local link updates, FFBS moves, auxiliary field Gibbs steps, and permutation updates leaves the joint measure $\pi(\mathcal{K}_\uparrow, \mathcal{K}_\downarrow, \Sigma, P_\uparrow, P_\downarrow)$ from Eq. 13 invariant.

*Proof Sketch*: Each component of the MCMC sweep satisfies detailed balance with respect to $\pi$: (1) Local updates use symmetric proposals with exact Metropolis-Hastings acceptance ratios derived from the transfer kernel. (2) The Forward-Filtering Backward-Sampling (FFBS) algorithm is an exact conditional sampler by construction, drawing from $P(\text{block}|\text{endpoints})$. (3) Auxiliary field updates are exact Gibbs steps that draw from the true closed-form conditional distributions. (4) Permutation updates use symmetric proposals with the appropriate acceptance probability to account for the sign change.

A composition of transition kernels that each satisfy detailed balance preserves the invariant distribution. $\square$

### A.2 Variance Analysis and Optimal Importance Sampling

When the joint measure $\pi$ is not strictly positive (the sign problem), we rely on importance sampling. The statistical variance of any observable is critically dependent on the choice of the positive reference distribution $q(x)$ from which we sample. Our reweighting scheme follows optimal importance sampling theory.

**Theorem (Optimal Variance)**: Among all positive reference distributions $q(x)$, the choice $q^*(x) \propto |w(x)|$, where $w(x)$ is the true (potentially signed) weight, minimizes the variance of the ratio estimator for any observable $\langle \mathcal{O} \rangle$.

*Proof Sketch*: This is a classic result from statistics. By the Cauchy-Schwarz inequality, the variance of the estimator is minimized when the sampling distribution is proportional to the magnitude of the numerator in the importance sampling formula. For an unbiased estimate of the expectation value, this corresponds to sampling from the absolute value of the original weight, $q^*(x) \propto |w(x)|$. $\square$

The resulting variance of an observable scales with the average sign, $\langle s \rangle_{\text{avg}}$, as $\sigma^2 \propto (\langle s \rangle_{\text{avg}}^2 N_{\text{samples}})^{-1}$. This inverse-square scaling of variance with the average sign is a fundamental limitation of any classical algorithm relying on importance sampling to overcome a sign problem.

### A.3 Trotter Error Analysis

The symmetric splitting of the evolution operator (Eq. 5) is the only source of systematic, controllable bias in the algorithm. This discretization introduces an error of $O(\Delta\tau^3)$ per time slice, which accumulates to a total error of $O(L_\tau \Delta\tau^3) = O(\beta \Delta\tau^2)$ for a fixed inverse temperature $\beta$. This means the method is second-order accurate.

This controlled error allows for systematic extrapolation to the "continuum limit," i.e., the true result of the time-continuous model, by performing simulations at several values of $\Delta\tau$ and extrapolating to $\Delta\tau \to 0$. For fixed $\beta$, halving the time step $\Delta\tau$ reduces the systematic error by a factor of four.

## B Background: Quantum Many-Body Physics and the Hubbard Model

### B.1 Historical Context and Importance of the Hubbard Model

The Hubbard model, introduced by John Hubbard in 1963 (Hubbard, 1963), represents the paradigmatic benchmark for strongly correlated electron systems in condensed matter physics. Originally proposed to describe metal-insulator transitions in transition metal oxides, it has become the minimal model for understanding complex quantum many-body phenomena where conventional perturbative approaches fail.

The model's computational significance stems from its deceptive simplicity masking extraordinary complexity. Despite involving only nearest-neighbor hopping and on-site repulsion, it captures the essential physics of diverse materials from cuprate superconductors to ultracold atomic gases. The two-dimensional case is particularly important as it is widely believed to contain the physics underlying high-temperature superconductivity (Anderson, 1987), discovered in cuprates in 1986. The model exhibits rich quantum phase diagrams with transitions between metallic, insulating, antiferromagnetic, and potentially superconducting phases driven by quantum fluctuations rather than thermal effects. These strongly correlated regimes give rise to exotic phenomena such as Mott insulators and non-Fermi liquids, making the Hubbard model both a fundamental theoretical challenge and a crucial testbed for computational methods in quantum many-body physics.

### B.2 QUANTUM FIELD THEORY FOUNDATIONS

To understand the computational challenges of the Hubbard model, we must first establish the quantum mechanical framework in which it operates.

#### B.2.1 QUANTUM STATES AND OPERATORS

In quantum mechanics, the state of a system is described by a state vector $|\psi\rangle$ in a Hilbert space. For many-body systems, this state vector encodes the quantum amplitudes for all possible configurations of particles. The complexity arises because the dimension of this Hilbert space grows exponentially with the number of particles.

Creation and Annihilation Operators: The Hubbard model is formulated using second-quantized operators. The creation operator $c_{r\sigma}^{\dagger}$ adds an electron with spin $\sigma$ at site $r$, while the annihilation operator $c_{r\sigma}$ removes one. These operators satisfy canonical anticommutation relations:

$$\{c_{r\sigma}, c_{r'\sigma'}^{\dagger}\} = \delta_{rr'}\delta_{\sigma\sigma'} \tag{24}$$

$$\{c_{r\sigma}, c_{r'\sigma'}\} = \{c_{r\sigma}^{\dagger}, c_{r'\sigma'}^{\dagger}\} = 0 \tag{25}$$

where $\{A, B\} = AB + BA$ denotes the anticommutator.

Occupation Number Representation: The many-body state can be expressed in the occupation number basis $|n_{1\uparrow}, n_{1\downarrow}, n_{2\uparrow}, n_{2\downarrow}, \ldots\rangle$, where $n_{r\sigma} \in \{0, 1\}$ indicates the presence or absence of an electron with spin $\sigma$ at site $r$.

#### B.2.2 FERMIONIC STATISTICS AND THE PAULI EXCLUSION PRINCIPLE

Electrons are fermions, particles with half-integer spin that obey the Pauli exclusion principle: no two fermions can occupy the same quantum state. This principle has profound consequences for the many-body wavefunction.

Antisymmetry Requirement: The many-body wavefunction must be antisymmetric under the exchange of any two fermions:

$$\psi(x_1, x_2, \ldots, x_i, \ldots, x_j, \ldots, x_N) = -\psi(x_1, x_2, \ldots, x_j, \ldots, x_i, \ldots, x_N) \tag{26}$$

where $x_i$ represents the combined spatial and spin coordinates of particle $i$.

Slater Determinants: For non-interacting fermions, the ground state wavefunction is a Slater determinant—a determinant of single-particle wavefunctions. This ensures proper antisymmetrization but becomes computationally intractable for interacting systems.

Sign Problem Origin: The antisymmetry requirement leads to quantum interference effects that can cause Monte Carlo sampling to suffer from exponentially small signal-to-noise ratios in certain parameter regimes.

### B.3 DETAILED ANALYSIS OF THE HUBBARD HAMILTONIAN

The full Hubbard Hamiltonian on a $d$-dimensional lattice is:

$$H = -t \sum_{\langle r,r'\rangle,\sigma} c_{r\sigma}^{\dagger} c_{r'\sigma} + U \sum_r n_{r\uparrow} n_{r\downarrow} - \mu \sum_{r,\sigma} n_{r\sigma} \tag{27}$$

where we have included a chemical potential term $\mu$ that controls the average particle density.

### B.3.1 PHYSICAL INTERPRETATION OF PARAMETERS

Hopping Parameter ($t$): Sets the kinetic energy scale. Larger $t$ favors delocalized, metallic behavior where electrons can move freely through the lattice. In real materials, $t$ is determined by the overlap of atomic orbitals between neighboring sites.

On-Site Repulsion ($U$): Represents the Coulomb repulsion between electrons occupying the same atomic orbital. Large $U$ favors localized, insulating behavior where double occupancy is suppressed. The competition between $t$ and $U$ drives the rich physics of the model.

Chemical Potential ($\mu$): Controls the average electron density $\langle n \rangle = \langle \sum_{r,\sigma} n_{r\sigma} \rangle / (2N)$, where the factor of 2 accounts for spin degeneracy. At half-filling, $\langle n \rangle = 1$ (one electron per site on average).

### B.4 HUBBARD-STRATONOVICH TRANSFORMATION: MATHEMATICAL DETAILS

The Hubbard-Stratonovich (HS) transformation is a powerful mathematical technique that converts interaction terms into auxiliary field formulations, enabling Monte Carlo sampling.

### B.4.1 GENERAL PRINCIPLE

The transformation is based on integral representations of exponentials. For a general quadratic form $\mathbf{x}^T A \mathbf{x}$, we have:

$$e^{\mathbf{x}^T A \mathbf{x}} = \frac{1}{\sqrt{\det(2\pi A^{-1})}} \int d\mathbf{s} \, e^{-\frac{1}{2}\mathbf{s}^T A^{-1} \mathbf{s} + \mathbf{s}^T \mathbf{x}} \tag{28}$$

This replaces a deterministic interaction term with a stochastic integral over auxiliary fields $\mathbf{s}$.

### B.4.2 DISCRETE HIRSCH TRANSFORMATION

For the repulsive Hubbard interaction, Hirsch (Hirsch, 1983) introduced a discrete auxiliary field transformation:

$$e^{-\Delta\tau U(n_{r\uparrow} - \frac{1}{2})(n_{r\downarrow} - \frac{1}{2})} = \frac{1}{2} \sum_{s_r = \pm 1} \cosh(\lambda) e^{-\lambda s_r (n_{r\uparrow} - n_{r\downarrow})} \tag{29}$$

where $\cosh(\lambda) = e^{\Delta\tau U/2}$ and $\lambda = \operatorname{arccosh}(e^{\Delta\tau U/2})$.

The key insight is that this transformation decouples the up and down spin interactions through the auxiliary Ising field $s_r$. Instead of the original quartic interaction $n_{r\uparrow} n_{r\downarrow}$, we now have linear couplings of each spin species to the auxiliary field.

### B.4.3 CONTINUOUS GAUSSIAN TRANSFORMATION

For attractive interactions ($U < 0$), a continuous Gaussian auxiliary field is more natural:

$$e^{|U|\Delta\tau n_{r\uparrow} n_{r\downarrow}} = e^{|U|\Delta\tau/8} \int \frac{d\sigma_r}{\sqrt{2\pi}} e^{-\sigma_r^2/2 + \sqrt{|U|\Delta\tau/2}\sigma_r (n_{r\uparrow} + n_{r\downarrow} - 1)} \tag{30}$$

This transformation has the advantage that the resulting weights are real and positive, eliminating sign problems for attractive systems.

### B.4.4 CONSEQUENCES FOR MONTE CARLO SAMPLING

After the HS transformation, the partition function becomes:

$$Z = \sum_{\{\sigma\}} P_0(\{\sigma\}) \langle 0| \prod_{\ell} e^{-\Delta\tau H_\ell(\sigma_\ell)} |0\rangle \tag{31}$$

where $P_0(\{\sigma\})$ is the auxiliary field prior and $H_\ell(\sigma_\ell)$ is the effective single-particle Hamiltonian at time slice $\ell$.

The fermion trace can be evaluated exactly, leading to determinantal weights. However, this requires $O(N^3)$ operations for each Monte Carlo update, creating the computational bottleneck that our method aims to overcome.

### B.5 Path Integral Formulation and Imaginary Time

The path integral approach to quantum mechanics, developed by Feynman (Feynman, 1948), provides the foundation for our worldline representation.

#### B.5.1 Wick Rotation and the Emergence of Statistical Mechanics

The Wick rotation represents one of the most profound connections between quantum mechanics and statistical mechanics. This transformation reveals why quantum systems at finite temperature can be studied using the same mathematical framework as classical energy-based models.

**From Oscillatory to Exponential Weights**. In real-time quantum mechanics, the time evolution operator is $e^{-iHt/\hbar}$, where $H$ is the Hamiltonian and $t$ is real time. The corresponding action in the path integral has the form $S = \int L \, dt$, where $L$ is the Lagrangian. This leads to path integral weights of the form $e^{iS/\hbar}$, which are oscillatory and difficult to evaluate numerically due to rapid phase cancellations.

The Wick rotation involves the analytic continuation $t \to -i\tau$, where $\tau$ is real and called "imaginary time" or "Euclidean time." Under this transformation:

$$e^{-iHt} \to e^{-H\tau} \tag{32}$$

$$e^{iS/\hbar} \to e^{-S_E/\hbar} \tag{33}$$

where $S_E$ is the Euclidean action. The oscillatory weights become exponentially decaying weights, which are much more amenable to numerical evaluation and Monte Carlo sampling.

**Connection to Thermal Equilibrium**. For systems in thermal equilibrium at temperature $T = 1/(\beta k_B)$, the density matrix is:

$$\rho = \frac{e^{-\beta H}}{Z}, \quad Z = \text{Tr}[e^{-\beta H}] \tag{34}$$

Comparing this with the Euclidean evolution operator $e^{-H\tau}$, we see that thermal equilibrium corresponds to Euclidean time evolution with $\tau = \beta$. This is the fundamental reason why finite-temperature quantum mechanics can be formulated as a path integral in imaginary time.

**Equivalence to Classical Statistical Mechanics**. After Wick rotation, the quantum partition function takes the form:

$$Z = \text{Tr}[e^{-\beta H}] = \sum_{\text{configurations}} e^{-\beta E(\text{configuration})} \tag{35}$$

This is precisely the same mathematical structure as classical statistical mechanics.

#### B.5.2 Trotter-Suzuki Decomposition

To make the path integral computationally tractable, we discretize imaginary time using the Trotter-Suzuki decomposition:

$$e^{-\beta H} = e^{-\Delta\tau H} \cdots e^{-\Delta\tau H} + O(\Delta\tau^2) \tag{36}$$

where $\beta = L_\tau \Delta\tau$ and we have $L_\tau$ time slices.

For the Hubbard model, we further decompose each time-slice evolution using symmetric splitting:

$$e^{-\Delta\tau(K+V)} = e^{-\Delta\tau K/2} e^{-\Delta\tau V} e^{-\Delta\tau K/2} + O(\Delta\tau^3) \tag{37}$$

This ensures that the discretization error is second-order in $\Delta\tau$, providing better convergence properties than first-order schemes.

#### B.5.3 Worldline Representation and Decoupling

In the discretized path integral, each particle traces out a worldline—a trajectory through imaginary time. For the Hubbard model, these worldlines live in momentum space and describe how the particle's momentum evolves from time slice to time slice.

**Worldlines as Markov Chains**. A worldline can be viewed as a discrete-time Markov chain in momentum space. For a single particle with spin $\sigma$, the worldline is a sequence:

$$\mathcal{W}_\sigma^{(p)} = \{k_0^{(p)}, k_1^{(p)}, k_2^{(p)}, \ldots, k_{L_\tau-1}^{(p)}, k_{L_\tau}^{(p)}\} \tag{38}$$

where $k_\ell^{(p)}$ is the momentum of particle $p$ at imaginary time slice $\ell$, and the superscript $(p)$ labels different particles.

The probability weight of this worldline is the product of single-step transition probabilities:

$$P(\mathcal{W}_\sigma^{(p)}) = \prod_{\ell=0}^{L_\tau-1} T_{\ell,\sigma}(k_{\ell+1}^{(p)} | k_\ell^{(p)}) \tag{39}$$

where $T_{\ell,\sigma}(k'|k)$ is the transition kernel from momentum $k$ to $k'$ during time slice $\ell$.

**Effect of Auxiliary Field Decoupling**. Before introducing auxiliary fields, the interacting fermion problem involves a complex many-body Hamiltonian where all particles are coupled through the interaction term $U \sum_r n_{r\uparrow} n_{r\downarrow}$. This coupling makes direct sampling of worldlines impossible because the transition probability for one particle depends on the positions of all other particles.

The Hubbard-Stratonovich transformation fundamentally changes this structure. After decoupling, the effective Hamiltonian becomes:

$$H_{\text{eff}}[\Sigma] = \sum_{\sigma,p} h_\sigma^{(p)}[\Sigma] \tag{40}$$

where $h_\sigma^{(p)}[\Sigma]$ is a single-particle Hamiltonian that depends only on the auxiliary field configuration $\Sigma$, not on other fermion positions.

This decoupling has profound consequences:

1. Independent Worldlines: Given a fixed auxiliary field configuration $\Sigma$, each fermion worldline evolves independently according to its own single-particle Hamiltonian.

2. Factorized Weights: The total weight of a fermion configuration factorizes as:

$$W(\{\mathcal{W}\}|\Sigma) = \prod_{\sigma,p} W_\sigma(\mathcal{W}_\sigma^{(p)}|\Sigma) \tag{41}$$

where each factor depends only on a single worldline.

3. Local Transition Kernels: The transition kernel for each worldline depends only on the auxiliary field at the corresponding spacetime locations, not on other fermion worldlines.

**Single-Particle Transfer Matrix**. For a given auxiliary field configuration $\Sigma$, the single-particle problem reduces to diagonalizing a transfer matrix. In momentum space, the effective single-particle Hamiltonian has the form:

$$h_\sigma[\Sigma] = K + V_\sigma[\Sigma] \tag{42}$$

where $K$ is the kinetic energy operator and $V_\sigma[\Sigma]$ is the effective potential created by the auxiliary fields.

The key insight is that $K$ is diagonal in momentum space while $V_\sigma[\Sigma]$ is diagonal in position space. After time slicing, the transfer matrix element becomes:

$$\langle k'|e^{-\Delta\tau h_\sigma[\Sigma]}|k\rangle = \langle k'|e^{-\frac{\Delta\tau}{2}K}e^{-\Delta\tau V_\sigma[\Sigma]}e^{-\frac{\Delta\tau}{2}K}|k\rangle \tag{43}$$

This is precisely the single-slice transfer kernel derived in the main text, which has the convolution structure that enables FFT acceleration.

**Fermionic Boundary Conditions and Permutations**. While auxiliary field decoupling makes individual worldlines independent, fermionic antisymmetry still requires special treatment at the boundaries. In the imaginary-time path integral with periodic boundary conditions in time, worldlines must form closed loops from $\tau = 0$ to $\tau = \beta$.

For fermions, these loops can be connected in different ways through permutations: (1) Identity Permutation: Each worldline closes on itself: $k_{L_\tau}^{(p)} = k_0^{(p)}$ (2) Pairwise Swaps: Worldlines exchange endpoints: $k_{L_\tau}^{(p)} = k_0^{(q)}$ and $k_{L_\tau}^{(q)} = k_0^{(p)}$ (3) Longer Cycles: Multiple worldlines form permutation cycles

Each permutation $P$ contributes a sign factor $\text{sgn}(P) = \pm 1$ to the total weight, which is essential for enforcing fermionic antisymmetry.

**Computational Advantage of Decoupling**. The decoupling achieved by auxiliary fields transforms the computational problem in several crucial ways: (1) Parallel Updates: Since worldlines are independent given $\Sigma$, they can be updated in parallel across different particles. (2) Efficient Block Sampling: The single-particle structure enables sophisticated sampling techniques like forward-filtering backward-sampling (FFBS) that would be impossible for interacting worldlines. (3) FFT Acceleration: The convolution structure of the single-particle transfer kernel enables $O(N \log N)$ updates via FFT, replacing the $O(N^3)$ matrix operations needed for the original interacting problem. (4) Exact Conditionals: The auxiliary fields have simple, factorized conditional distributions that can be sampled exactly, enabling efficient Gibbs sampling steps.

**Comparison with Traditional DQMC**. Traditional DQMC also uses auxiliary field decoupling but then integrates out the fermion worldlines analytically. This leads to:

$$Z = \sum_\Sigma P(\Sigma) \prod_\sigma \det[M_\sigma(\Sigma)] \tag{44}$$

While this eliminates the need to sample worldlines explicitly, it creates the determinant bottleneck that scales as $O(N^3)$. Our approach instead samples both worldlines and auxiliary fields jointly, avoiding determinants entirely while maintaining the computational advantages of decoupling.

