# OpenReview forum: "FFT-Accelerated Auxiliary Variable MCMC for Fermionic Lattice Models: A Determinant-Free Approach with $O(N \log N)$ Complexity"
_ICLR.cc/2026/Conference — ICLR 2026 Conference Withdrawn Submission_

### Official Review · Reviewer_TPrB · 2025-10-17

**Soundness:** 1
**Presentation:** 3
**Contribution:** 3
**Rating:** 2
**Confidence:** 4

**Summary:**

The paper introduces a method for solving Hubbard-type models by calculating the proposal probability using the FFT. They apply the algorithm to the Hubbard model and compare the numerical results with analytic results in certain limits.

**Strengths:**

Simulating fermionic problems with larger system size is an important task that the paper is approaching.
I appreciate the comparison of the numerical results to the known analytic results to validate the approach.

**Weaknesses:**

My main concern is that the main claim of the paper, namely that this approach has an O(V log V) scaling across two orders of magnitude, is just stated, but the experiments to support this claim are not shown. Also, the statement that they achieved a 5 - 10 fold improvement over the state-of-the-art is just stated, no numbers are shown. Details about the experiments are also lacking, for example, are the methods compared at the same accuracy, i.e. the computational cost for a fixed accuracy (or equivalently for the fixed effective sample size) should be shown. The paper also states that its results are consistent with the established DQMC results, but they do not show this in plots. I am willing to increase my score if the authors provide evidence for their claims.

**Questions:**

Apart from the main concern, I would appreciate some other clarifications:

1. In Lattice quantum chromodynamics, the determinant problem is typically solved by the pseudo fermion fields that avoid computing the determinant in favor of solving a linear system (with conjugate gradient) and is combined with Hamiltonian Monte Carlo sampling. This approach has also been used in solid state physics (e.g. https://www.nature.com/articles/s41467-023-37686-4) for problems similar to the the Hubbard model in this paper. Can authors comment on the HMC approach and compare it with their work?

2. As far as I know, the local updates change the matrix in the determinant by a low rank, so the Sherman–Morrison formula can be used to make the determinant update cheaper than O(V^3)?

---

### Official Review · Reviewer_KCZE · 2025-10-31

**Soundness:** 2
**Presentation:** 1
**Contribution:** 2
**Rating:** 2
**Confidence:** 1

**Summary:**

This paper introduced a auxiliary-variable MCMC approach to simulate quantum many-body problems. They further leverages the
Fast Fourier Transform (FFT) to implement the block-sampling updates (FFBS) with a complexity of around O(N logN). The method is validated on 1D and 2D Hubbard models, cornerstone problems in condensed matter physics.

**Strengths:**

+ The FFT implementation can greatly reduce the computation complexity of the transition kernels.
+ The proposed method shows 10× speedup compared with the DQMC implementations.

**Weaknesses:**

The reviewer has a few concerns about the weakness of this work:
+ The paper scope is focused on quantum simulation, rather than machine learning. As a result, this paper may have very limited benefit to the machine learning community. Journals in computational physics or quantum physics may be a better publication venue for this work.
+ This work only compared with one basline method DQMC, and this baseline was very old (in 2012). More baseline methods should be compared to provide a fair and comprehensive evaluation.
+ This work didn't show any application examples related to machine learning, so it's not clear how this work can advance the frontier of machine learning.
+The paper writing is very quantum-physics oriented, and the contents are hard to understand for generic machine learning researchers.

**Questions:**

1. Could you please clarify how this method could benefit machine learning topics/tasks?
2. Can you provide a comparison with more basline methods and with more details (e.g., memory cost, parallelism efficiency, computing flops, etc)?

---

### Official Review · Reviewer_ZoHE · 2025-11-03

**Soundness:** 4
**Presentation:** 2
**Contribution:** 3
**Rating:** 4
**Confidence:** 2

**Summary:**

This work proposes a Gibbs sampler for fermionic lattice models to circumvent the cubic cost of conventional approaches which rely on the marginalisation of latent variables.

**Strengths:**

This work seems to address an important computational limitation in the MCMC sampling of fermionic lattice models. I have no reason to doubt the formal validity of the proposed methodolgy (although, admittedly, I did not follow quite a few parts of it). The main idea, i.e., using an extended state-space construction paired with Gibbs sampling to circumvent costly marginalisation, seems sensible in this context.

**Weaknesses:**

**Clarity:**
This work seems written for an audience well familiar with the notational conventions used in the literature on quantum simulation. It is therefore not sufficiently self contained. This may be suitable for a physics journal but I don't think this work is accessible to the typical ICLR audience. For instance.

1. Quite a few symbols (e.g., angle-bracket notation, $\mathcal{D}$, $S_{\max}$, $\beta$, $E$, $N$) are not defined (or at least, not defined upon first use). I can infer the meaning of some of these symbols from context or with a bit of additional reading. But I expect the average reader won't and will therefore give this work a miss.

2. Following the specification of the model and methodology seems to require a deep understanding of the underlying physics. However, the main contribution of this paper seems to be the exploitation of a standard extended state-space/auxiliary variable trick to exploit costly marginalisation which is very common in machine learning/computational statistics. Thus, for an ICLR audience, I think it would be much better to provide a more "high-level" description of the methods which abstract away the concepts, terminology and conventions from the physics literature.

**Questions:**

1. Gibbs samplers like the one proposed here will break down if latent variables that are updated separately are highly correlated. Thus, there will likely be regimes (parameter values) in which such a Gibbs sampler performs worse than the marginalised sampler, even when taking computational complexity into account (because the Gibbs sampler can have an arbitrarily bad convergence rate in the presence of such correlations). Thus, my question is, is there a model regime in which this approach fails? And how realistic is such a regime in real-world applications?

2. Why was this submitted to ICLR? Wouldn't this be better placed in a suitable physics journal?

---

### Official Review · Reviewer_kGDS · 2025-11-03

**Soundness:** 2
**Presentation:** 2
**Contribution:** 1
**Rating:** 0
**Confidence:** 2

**Summary:**

This paper proposes a method for sampling fermionic lattice models based on a Hubbard-Stratonovic decoupling of the Fermionic interaction. This is equivalent to introduce an auxiliary-field which mediate the interaction between Fermions and leading to a Gibbs sampling method which alternate between Fermions and the latent field, inducing a reduction in scaling complexity from cubic to $Nlog(N)$ with system size compared to methods based on Slater determinant representations.

**Strengths:**

The introduction of the latent field yields a much more more efficient MC scheme compared to some other method based on Slater determinant representations.

**Weaknesses:**

Auxiliary-field quantum Monte Carlo seems not to be new in this domain of correlated Fermions (see https://journals.aps.org/prx/pdf/10.1103/PhysRevX.5.041041 and refs inside) and it is a technique among many others, which is limited to computing the ground state properties i.e. the static properties of the Hubbard model. Calling this Machine learning approach is quite misleading because here no representation is actually learned, and Hubbard-Stratonovic  transform is known for a long time (prior to ML actually). For instance in the context of NN  the Hopfield model can be reformulated in terms of a binary-Gaussian restricted Boltzmann machine thanks to a Hubbard-Stratonovic transformation,  but ML concerns the actual learning of these models which are otherwise equivalent. Hence to me this paper is not of general interest to the ML community, it tackle a very specialized problem of condensed matter physics with methods which are not ML in particular, in the sense that HS, FFt or matrix multiplication is not specific to ML.

**Questions:**

1- How this work compares to techniques describes in https://journals.aps.org/prx/pdf/10.1103/PhysRevX.5.041041 ?
2- What new results concerning the physics of the Hubbard model could be expected to be obtained with this technique?
3- could the method be extended to the dynamics?

In my point of view, without judging the scientific contribution of this work (c.f. my reserve concerning novelty), I believe
this work should be submitted to specialized physics methods journals rather than general machine learning conferences or journals.

---

### Public Comment · ~Lei_Wang2 · 2025-11-24
**sign problem of the approach**

Interesting contribution!

Is the joint probability Eq. (13) of the manuscript always positive for the usual DQMC sign free cases (say, attractive Hubbard model, or repulsive Hubbard model on bipartite lattice at half-filling) ?

If so, it would be nice to have a detailed proof.
If not, it will be useful to analyze the severeness of the sign problem.

---

### Note · Authors · 2025-12-11

**Comment:**

We would like to formally withdraw this submission. After further discussion, we decided to prepare a substantially revised version aimed at a physics journal, where the methodological and theoretical contributions are a better fit. We appreciate the reviewers’ time and feedback.

**Withdrawal Confirmation:**

I have read and agree with the venue's withdrawal policy on behalf of myself and my co-authors.